# Influences of Ultrasonic Treatments on the Structure and Antioxidant Properties of Sugar Beet Pectin

**DOI:** 10.3390/foods12051020

**Published:** 2023-02-28

**Authors:** Yingjie Xu, Jian Zhang, Jinmeng He, Ting Liu, Xiaobing Guo

**Affiliations:** Food College, Shihezi University, Shihezi 832003, China

**Keywords:** pectin, ultrasonication, structure, antioxidant properties

## Abstract

The objective of this study was to explore the structural changes and oxidation resistance of ultrasonic degradation products of sugar beet pectin (SBP). The changes in the structures and antioxidant activity between SBP and its degradation products were compared. As the ultrasonic treatment time increased, the content of α-D-1,4-galacturonic acid (GalA) also increased, to 68.28%. In addition, the neutral sugar (NS) content, esterification degree (DE), particle size, intrinsic viscosity and viscosity-average molecular weight (*M_V_*) of the modified SBP decreased. Fourier transform infrared (FT-IR) spectroscopy and scanning electron microscopy (SEM) were used to study the degradation of the SBP structure after ultrasonication. After ultrasonic treatment, the DPPH and ABTS free radical scavenging activities of the modified SBP reached 67.84% and 54.67% at the concentration of 4 mg/mL, respectively, and the thermal stability of modified SBP was also improved. All of the results indicate that the ultrasonic technology is an environmentally friendly, simple, and effective strategy to improve the antioxidant capacity of SBP.

## 1. Introduction

Pectin is a multifunctional hetero-polysaccharide found in plant cell walls that consists of a series of acidic polymers and partially neutral polymers [1]. The pectic polysaccharide molecule is composed of a linear homogalacturonan main chain consisting of GalA and multi-branched sidechains of arabinogalactan and rhamnogalacturonan. The functionality of pectin depends largely on its molecular weight [2], methyl esterification degree [3], chain-to-chain interactions, protein content [4] and side-chain structure. Due to the complexity of the fine structure of the pectin side chain, the difference in NS type and content affect the functional characteristics of pectin [5,6,7]. In addition, the source of raw materials and different extraction methods of pectin also lead to different functional characteristics and applications [8,9].

Pectin can be divided into high-methoxy pectin and low-methoxy pectin according to the methoxy content. Pectin with different degrees of esterification has different applications. The consumption of low-fat foods is a current trend, and high-fat foods have gradually become an unpopular choice among consumers. For example, low-methoxyl pectin-based fat mimetics could be added in mayonnaise to reduce fat addition, and there is no obvious difference in sensory evaluation compared with full-fat mayonnaise [10]. High-methoxy pectin is often used in acidic milk beverages to stabilize the dispersion due to its better emulsification and gel properties [11]. In addition, the antioxidant properties of pectin have also attracted attention in recent years. Previous study has shown that pectin samples extracted from *Opuntia robusta* via microwave and conventional heating methods presented different structural characteristics and antioxidant capacity. The pectin sample with lower molecular weight and higher GalA content showed better DPPH radical scavenging capacity due to the exposure of more active sites [12]. 

Ultrasound is a type of mechanical wave that produces bubbles upon passing through a solution. The shock waves generated by bubble collapse can directly impact particles in solution, and the shear force can cause fragmentation [13]. When the frequency is within a certain range, the bubbles produced by the ultrasound increase with the increase in power [14]. Ultrasonic action will also produce free radicals, and redox reactions occur with solutes [15]. The application of ultrasonic technology in the food industry is becoming more and more popular. Compared with other methods, ultrasonication is an environmentally friendly and economical choice. Currently, ultrasonication has been mostly applied to modify high-molecular-weight substances (such as protein and polysaccharide) to improve their functional properties. In a recent study, the emulsifying and foaming properties of black bean protein isolates were improved after ultrasound treatment due to their structural changes [16] Many studies have shown that ultrasound treatment can reduce the molecular weight of polysaccharides and improve their biological activity. For example, *Porphyra yezoensis* polysaccharides modified by ultrasonic treatment can effectively inhibit the proliferation of SGC7901 cells. The reason is probably due to the fact that the reduction of the molecular weight of polysaccharides can enhance the interaction between certain functional groups of molecules [17]. After ultrasound treatment, the viscosity of high-methoxy pectin decreased, and its gel ability became weaker [18]. Ultrasound treatment enhanced the film-forming properties of amylose by increasing its solubility and decreasing its viscosity [19]. The antioxidant activity of sweet potato pectin was improved by ultrasonication, which might be due to the exposure of hidden active sites in the pectin chain by ultrasonic treatment [20]. In addition, ultrasonication could improve the solubility and biological activity of pectin to a certain extent. However, excessive ultrasonic treatment can be detrimental to its functional performance [21,22]. Therefore, it is very necessary to optimize the treatment conditions according to different raw materials and applications.

As far as we know, there is no report on the effect of ultrasonic treatment on SBP and its antioxidant capacity. Based on the above analysis, we anticipated that ultrasonic technology may improve the antioxidant properties of SBP. Ultrasonication was used to modify SBP structure under different treatment conditions. The effects of ultrasonic modification on the SBP’s chemical composition (GalA/Gal/Rha/Ara/Glc/protein), intrinsic viscosity, particle size, color, microstructure, thermal stability and antioxidant activity were evaluated. We hoped to improve the thermal stability and antioxidant capacity of SBP by changing its chemical structure via ultrasonic technology.

## 2. Materials and Methods

### 2.1. Materials

SBP was supplied by Azelis International Trade Co., Ltd. (Shanghai, China). GalA standard was purchased from Sigma (Sigma-Aldrich, St. Louis, MO, USA). Galactose (Gal), Rhamnose (Rha), Arabinose (Ara), Glucose (Glc), and other reagents were purchased from Macklin Reagent Co., Ltd. (Shanghai, China).

### 2.2. Ultrasonic Treatment of SBP

Two grams of SBP (native) were weighed and dissolved in 100 mL distilled water overnight at room temperature. An ultrasonic cell disruptor (SCIENTZ-IID, Ningbo Xinzhi Biological Technology Co., Ltd. Ningbo, China) equipped with a 0.6 cm diameter probe was used for ultrasonic treatments. The dissolved SBP was placed in a 250 mL high-type beaker for ultrasound treatment (20 kHz frequency) at the power of 500 W. The ultrasonic treatment times were respectively set as 10, 20, 30, 60, and 90 min. The SBP solution after ultrasonication was dialyzed (10,000 Da) for 48 h, and then lyophilized for later use. 

### 2.3. Chemical Composition

The GalA content of different SBP samples (with or without ultrasonic treatment) was determined by the *m*-hydroxydiphenyl method [23]. The neutral sugar content of different SBP samples was determined by 1-pheny-3-methyl-5-pyrazolone (PMP) pre-column derivatization [24]. The monosaccharide composition (Gal, Rha, Ara, Glc) of SBP and its degradation products were determined. Briefly, 5 mg of SBP was weighed into a digestion tube, and 2 mL of 2 M TFA was added for hydrolysis in an oven at 110 °C for 6 h. Then, the samples were put into an ice-water bath for cooling. After cooling, the pH was adjusted to pH 7.0 using 5 M NaOH, and then 400 μL of hydrolyzed sample was transferred into a test tube. After that, 400 μL of 0.5 M PMP methanol solution was added, followed by 400 μL of 0.3 M NaOH solution. After the mixture had been mixed thoroughly, it was transferred to a water bath at 70 °C for 30 min. When the mixed solution was cooled to room temperature, 400 μL of 0.3 M HCl was added to neutralize the NaOH. Finally, chloroform was added to remove excess PMP, and then the sample was passed through a 0.45 μm organic phase filtration membrane. The PMP derivatization of standard monosaccharides was carried out in the same way. Protein content was determined by the Kjeldahl method, and the protein conversion factor was 6.25. 

### 2.4. Esterification Degree (DE) 

The DE of different SBP samples was determined by a previous method with slight modification [25]. A 20 mL sample solution of SBP (1 mg/mL) was transferred into a conical flask, and then 200 μL of phenolphthalein was added to the solution of SBP as an indicator. An NaOH solution (0.05 M) was used to titrate the SBP solution. When the color of the SBP solution turned to light pink, the consumption volume of NaOH was recorded as *V*_1_. Subsequently, 10 mL of 0.1 M NaOH was added into that mixture and stirred continuously for 30 min. After that, the excess NaOH was neutralized by using 10 mL of 0.1 M HCl. Finally, 0.05 M NaOH was used to titrate that mixed solution of SBP until its color turned to light pink. The consumption volume of NaOH was recorded as *V*_2_. The DE was calculated as follows:(1)$$\mathrm{DE}\left(\%\right)=\frac{V_{2}}{V_{1}+V_{2}}\times 100$$

### 2.5. Intrinsic Viscosity and M_V_

The intrinsic viscosity of different SBP samples was measured by using an Ubbelohde capillary viscometer, according to the method reported by Diao et al. with slight modifications [26]. The SBP sample was dissolved in 0.1 M NaCl solution (through a 0.45 μm pore-size membrane filter) at a concentration of 1 mg/mL. The time (*t_s_*) for the solution to flow out of the capillary after placing 10 mL of the sample solution in the viscometer was recorded. The above procedure was repeated, replacing the pectin solution with 0.1 M NaCl, and the time (*t*_0_) was recorded. The intrinsic viscosity was calculated as follows: (2)$$\left[\eta \right]=\frac{\sqrt{2\;(\eta_{sp}-\ln\eta_{r})}}{c}$$
(3)$$\eta_{r}=\frac{t_{s}}{t_{0}}$$
(4)$$\eta_{sp}=\eta_{r}-1$$
where [*η*] is intrinsic viscosity; *η_r_* is relative viscosity; *η_sp_* is incremental viscosity; c is the concentration of SBP.

The relationship between intrinsic viscosity and viscosity-average molecular weight follows the Mark–Houwink Equation (5) [27].
(5)$$\left[\eta \right]=kW_{V}^{\alpha }$$
where [*η*] is the intrinsic viscosity (L/g), *M_V_* is the viscosity-average molecular weight, and the value of *k* and *α* are 2.34 × 10^−5^ and 0.8224, respectively [28].

### 2.6. Particle Size

The particle size was determined by the method reported by Chen et al. [29].The different SBP samples were fully dissolved in deionized water before the experiment. The concentration of sample solution was 1 mg/mL. The insoluble substance was removed by centrifugation at 10,000 rpm, and then the particle sizes of different SBP samples were determined using a Zeta potential analyzer (Malvern Zeta-sizer Nano, UK). All tests were performed at 25 °C.

### 2.7. Color Analysis 

A handheld colorimeter (SC-10, 3nh, Shenzhen, China) was used to determine the color of different SBP samples. The SBP sample was homogenized before the measurement, and then the colorimeter was used to determine the L*, a* and b* values of the sample. The value of L* is the brightness, in which 0 represents black and 100 represents white. The value of a* ranges from red to green, where the positive value is red, and the negative value is green. The b* value ranges from yellow to blue, where the positive value is yellow, and the negative value is blue.

A handheld colorimeter (SC-10, 3nh, Shenzhen, China) was used to determine the color of different SBP samples. The SBP sample was homogenized before the measurement, and then the colorimeter was used to determine the L*, a* and b* values of the sample. The value of L* is the brightness, in which 0 represents black and 100 represents white. The value of a* ranges from red to green, where the positive value is red, and the negative value is green. The b* value ranges from yellow to blue, where the positive value is yellow, and the negative value is blue.

### 2.8. FT-IR Analysis

SBP and KBr were mixed homogeneously in a ratio of about 1:100. Then, the mixture was pressed into a pellet. The analysis was performed using an FT-IR spectrometer (Bruker Vertex 70V, Germany). The scan range was set as 4000–400 cm^−1^.

### 2.9. SEM Analysis

The native SBP and the ultrasonically treated SBP were fixed on a sample holder with conductive tape. The samples were sputtered with gold powder under vacuum. Images were observed and recorded using a scanning electron microscope (su8010, Hitachi Co., Tokyo, Japan) at an accelerating voltage of 10 kV.

### 2.10. Thermal Analysis

With reference to a previous method [30], the thermal stability of different SBP samples was determined by TG (TGA/DSC 3+, METTLER TOLEDO, Switzerland) and DSC (209F1, NETZSCH, Germany) techniques. The TG measurement temperature range was 40–600 °C. The DSC measurement temperature range was 50–300 °C. The heating rate was 10 °C/min, and the carrier gas was nitrogen. Five milligrams of the sample were weighed in a crucible, using the empty crucible as a reference.

### 2.11. Antioxidant Capacity Analysis

#### 2.11.1. DPPH Radical Scavenging Activity

The DPPH radical scavenging activity of SBP was determined by the method reported by Chen et al. [29]. A 0.2 mM DPPH methanol solution and a series of SBP solutions with different concentrations were prepared in advance. Two milliliters of SBP solution were mixed with 2 mL DPPH methanol solution, followed by reacting in the dark for 30 min. The absorbance was measured at 517 nm.
(6)$$\mathrm{DPPH}\;\mathrm{radical}\;\mathrm{scavenging}\;\mathrm{activity}\;\left(\%\right)=(1-\frac{A_{i}-A_{j}}{A_{0}})\times 100$$
where *A_i_* is the absorbance of the DPPH methanol + sample, *A_j_* is the absorbance of the methanol + sample, and *A*_0_ is the absorbance of the DPPH methanol + methanol.

#### 2.11.2. ABTS Radical Scavenging Activity

A previously reported method was used to evaluate the ABTS radical scavenging activity of different SBP samples with slight modification [31]. A 5 mL solution of 7 mM ABTS solution was prepared, followed by mixing with 1 mL of 15 mM potassium persulfate. Then, the mixed solution was incubated in the dark for 16–20 h at room temperature. The ABTS solution was diluted to achieve an absorbance of 0.7 ± 0.02 at 734 nm. Subsequently, 0.2 mL SBP solution was mixed with 2 mL of ABTS, and the absorbance was measured at 734 nm after standing for 10 min.
(7)$$\mathrm{ABTS}\;\mathrm{radical}\;\mathrm{scavenging}\;\mathrm{activity}\;\left(\%\right)=(1-\frac{A_{x}}{A_{0}})\times 100$$
where *A_x_* is the absorbance of the ABTS + sample, and *A*_0_ is the absorbance of ABTS + distilled water.

#### 2.11.3. Reducing Power

The reducing power of different SBP samples was determined via the method reported by Aiyegoro et al. [32]. First, 1 mL of pre-dissolved pectin solution was placed in a 20 mL glass-stoppered test tube, followed by the addition of 2.5 mL of 1% K_3_[Fe (CN)_6_] solution. The mixture was placed in a 50 °C water bath for 20 min, and then 2.5 mL 10% TCA was added after cooling to terminate the reaction. Then, 2.5 mL of the above mixture was removed and combined with 2.5 mL of distilled water and 1 mL of 0.1% FeCl_3_ solution. The solution was mixed evenly and allowed to stand for 10 min. The absorbance was measured at 700 nm.

### 2.12. Statistical Analysis

All experiments were performed in triplicate, and the data were presented as the means ± SD. Significant differences between means (*p* < 0.05) were determined through Tukey’s HSD test by one-way ANOVA using SPSS 26.0 (SPSS Inc., Chicago, IL, USA).

## 3. Results and Discussion

### 3.1. Structural Characterization

Table 1 shows that GalA was the main component that constituted the backbone of SBP, which indicates that SBP is an acidic polysaccharide. With the prolongation of ultrasonic treatment, the content of GalA increased from 63.68% to 68.28%, which might be because the NS side chains of SBP were more sensitive to ultrasound that was more easily damaged, while the main chain region was relatively stable. The same phenomenon occurred in the process of ultrasonic-assisted extraction of pectin from mango peel. With the increase in ultrasonic power, the content of GalA in pectin extracted from mango peel increased significantly. Ultrasonic waves may lead to the degradation of pectin side chain, thus increasing the GalA content of pectin [33]. With further prolongation of ultrasonication time, the increase in GalA content of SBP was not obvious, which might be due to the degradation of pectin backbone.

With regard to the NS content, it was reduced with the prolongation of the ultrasound treatment time. Rha and Gal are the main components of the pectin side chains. However, the effect of ultrasonic treatment on their content was not as obvious as that on other neutral saccharides. In contrast, ultrasonic treatment had a significant effect on Ara and Glc content, though the content of Glc in SBP polysaccharides was very small. At 90 min of ultrasonic treatment, the NS content decrease rate began to slow down. A possible explanation for this phenomenon was that the structure of SBP under this condition would not be further degraded, which would require higher ultrasonic intensity or other auxiliary methods to break. The results show that the types of monosaccharides did not change after ultrasonic treatment, but the content of NS was affected, which was similar to the results of Wang et al. They found that ultrasonic treatment had no significant effect on monosaccharide types of yellow tea polysaccharide [34]. In addition, the effect of ultrasound treatment on the protein (Pro) content of SBP was not obvious.

Pectin is mainly composed of GalA residues in which some of the C-6 carboxyl groups are methyl-esterified. DE is the ratio of the amount of esterified GalA to the total GalA content [35]. We found that the DE of the native SBP was 51.4%, which indicated that the SBP in this study was a type of pectin with a low degree of methyl esterification. From Table 1, the DE of SBP and ultrasonicated SBP were 49.75%, 47.7%, 44.9%, 42.78% and 41.5%, respectively. We could clearly see that the ultrasonic treatment reduced the DE of SBP, which may be due to the hydrolysis of ester groups by ultrasonic treatment, resulting in more free carboxyl groups exposed on SBP chains. Previous studies have showed that pectin with a low esterification degree contains more free carboxyl groups because when ultrasonication hydrolyzed the ester groups, the DE decreased [20].

### 3.2. Intrinsic Viscosity and M_V_

Intrinsic viscosity is the most common indicator used to represent the viscosity of polymer solutions. It can be seen from Equation (5) that intrinsic viscosity is positively correlated with molecular weight. Figure 1a shows the intrinsic viscosity and *M_V_* at different ultrasonic treatment times. The intrinsic viscosity significantly decreased when the ultrasonic treatment time was 10 min. As the ultrasonic treatment time increased, the degradation of SBP accelerated, and the chain structure of pectin deteriorated, leading to a decrease in intrinsic viscosity and *M_V_*. 

### 3.3. Particle Size 

The particle size of pectin is an important indicator to characterize whether or not pectin degradation occurred. As shown in Figure 1b, the particle size of SBP was significantly reduced from 879 nm to 489 nm after ultrasonic treatment. When the treatment time increased, the particle size of SBP gradually decreased. The same results were obtained in the research of hawthorn pectin. Considering that the pectin particle size is positively correlated with the molecular weight (r = 0.962, *p* < 0.05), the molecule of SBP was obviously degraded by ultrasonic treatment [29]. However, the rate of decrease in SBP particle size began to slow down with the extension of processing time, indicating that long-chain pectin was more easily degraded, while the degradation of short-chain pectin required higher ultrasonic intensity or longer ultrasonication time. The polydispersity index (PDI) is an indicator of the degree of homogeneity in the reaction system [36]. In this study, the PDI decreased gradually with the increasing power and time of ultrasonic treatment, indicating that the chain of the SBP molecule was destroyed by ultrasonic waves. The decreased PDI value indicated that the SBP sample after ultrasonic treatment was more homogeneous.

### 3.4. Color Analysis of SBP and Its Degradation Products

Table 2 shows that after ultrasonic treatment, the value of L* increased, a* decreased, and b* performance was not obvious. This phenomenon could be due to the following reasons. On the one hand, ultrasound destroyed the chromogenic groups of SBP. On the other hand, ultrasound destroyed some glycosidic bonds of SBP, which could unravel the tangled chains between pectin molecules and lead to intermolecular interaction changes. Long’s research showed that the free radical produced by H_2_O_2_−V_C_ destroyed the glycosidic bond, and the brightness of degraded polysaccharide was improved [37]. The results for color values showed that the chromogenic groups of SBP were affected by ultrasonic treatment. 

### 3.5. FT-IR

As shown in Figure 2, the FT-IR absorption peaks of different SBP samples were similar, indicating that the ultrasonic treatment did not have obvious effects on the primary structure and glycosidic bonds of SBP. The stretching vibration of O−H at 3402 cm^−1^ was caused by moisture in the pectin. The peaks at 2941 cm^−1^, 1441 cm^−1^, and 1102 cm^−1^ were the stretching vibrations of C−H, C−OH, and C−O, respectively [38]. The absorption peaks at 859 cm^−1^ and 515 cm^−1^ indicated the existence of α- and β-glycosidic bonds [39]. The intensity of the absorption peaks weakened with the prolongation of ultrasonication time, indicating that some glycosidic bonds of SBP were broken by ultrasonication. Peaks around 1100 cm^−1^ and 1200 cm^−1^ were attributed to R−O−R ether and C−C ring bonds present in the pectin ring structure. During ultrasonic treatment, the peak intensity weakened with increasing treatment time, indicating that the cyclic structure of SBP was destroyed [40]. The ratio of the sum of the peak area (−COO−R) of 1745 cm^−1^ to the peak area (−COO−R) of 1745 cm^−1^ and the peak area (COO−) of 1627 cm^−1^ was used to calculate the DE of pectin [41]. The peak areas of 1745 cm^−1^ and 1627 cm^−1^ decreased with the extension of ultrasonication time, indicating that the DE of pectin decreased after ultrasonic treatment, which was also proved by the above experimental results.

### 3.6. SEM Analysis

In Figure 3, the microstructure of control SBP showed an irregular sheet structure; the surface was smooth, and the edge was relatively flat. With increasing ultrasonic treatment time, the microcosmic surface of the SBP became rough, and irregular tearing and saw-tooth shapes appeared at the edge. These microstructural changes in SBP might be caused by the strong blasting and tearing force of bubbles generated by ultrasonic cavitation. Compared with the integrity of the native SBP, the SBP after ultrasonic treatment had many fragments. The increase in tiny fragments in SBP further indicated that the mechanical wave generated by ultrasonication degraded the pectin structure. 

### 3.7. Thermal Analysis

As can be seen from Figure 4, the TG/DTG data showed that all SBP samples had noticeable weight loss at 250 °C. That was due to the thermal degradation of SBP, which was related to its chemical composition. As shown in Figure 4, all SBP samples had an endothermic peak at about 110–120 °C, which was caused by the breakage of hydrogen bonds during water evaporation in the pectin sample [42]. When the temperature increased, a second exothermic peak appeared at about 250 °C, which was caused by the break of the glycosidic bonds in the pectin sample. As shown in Table 3, the T_m DSC_ (temperature of melting) of SBP decreased after ultrasonic treatment. From the changes in T_d DTG_ and T_d DSC_ (temperature of degradation), the thermal degradation temperature moved in the direction of increasing temperature, indicating that short-term ultrasonic treatment improved the thermal stability of pectin. In addition, the thermal degradation temperature of SBP began to decrease at 90 min. Those results were probably due to degradation of the pectin backbone after long-term ultrasonic treatment. Previous studies have shown that higher content of GalA in pectin means that more energy is needed to complete the transformation [43]. The decrease in molecular weight also affects the thermal stability of SBP. Chen’s study showed that after ultrasonic degradation, the molecular weight of polysaccharides decreased, the structure was loose, the intermolecular hydrogen bonding was enhanced, and the thermal stability was also enhanced [44]. Relevant results indicated that the ultrasonic treatment could improve the thermal stability of SBP.

### 3.8. In Vitro Antioxidant Activity of SBP

Figure 5a shows the DPPH radical scavenging activity of different SBP samples. The DPPH radical scavenging capacity of SBP was concentration-dependent. When the SBP concentration was 4 mg/mL, the DPPH radical scavenging capacity reached 67.84%. Under the same concentration condition, the SBP modified by ultrasonication had a higher DPPH radical scavenging ability than that of native pectin. The ABTS assay is the most widely used method of indirect detection for the determination of free radical scavenging capacity, and it also can be used for the determination of the antioxidant capacity of hydrophilic and lipophilic substances. Figure 5b shows the ABTS radical scavenging capacity of different SBP samples. When the ultrasonic treatment time was 30 min, the free radical scavenging capacity of SBP (4 mg/mL) reached 59.04%. However, the in vitro antioxidant activity of SBP decreased with increasing treatment time, which may be due to the fact that active structures of SBP were damaged by ultrasonic treatment. In a previous study, the antioxidant properties of different varieties of citrus pectin were studied. With regard to ABTS radical scavenging ability, low molecular weight pectin had a lower IC_50_ value and better antioxidant activity [45]. 

Figure 5c shows the reducing power of different SBP samples. Those data were related to the ability of antioxidants to provide electrons. For a given concentration of pectin, the reducing power of SBP first increased and then decreased with the prolongation of ultrasonic treatment time, indicating that moderate ultrasonic treatment conditions could significantly improve the reducing power of SBP. Previous studies also showed that pectin with lower molecular weight and higher GalA content would produce more reducing ends that were conducive to improving its reduction capacity [46,47]. Hydroxyl groups contained in polysaccharides have free radical scavenging ability [48]. A reduction in DE increased the number of free carboxyl groups upstream of the ultrasound-modified SBP chain structure and the accessible hydroxyl groups on the surface [49]. Moreover, the intrinsic viscosity decreased, and the solubility increased, which was beneficial to the exposure of active parts of SBP and the improvement of antioxidant activity. A similar conclusion was reached by Venzon et al. [50]. Therefore, these facts proved that ultrasonic modification is an effective way to significantly improve the reducing power of SBP.

Generally speaking, the antioxidant activity of SBP is positively correlated with the concentration. For a given concentration of SBP, ultrasonic treatment times that are too long or too short are not conducive to improving the antioxidant capacity of SBP. When the ultrasonic treatment time is too long, the active structures of SBP can be damaged by ultrasonic waves. However, if the ultrasonic treatment time is too short, the antioxidant active sites of SBP cannot be fully exposed. Therefore, ultrasonic treatment can be used to improve the antioxidant capacity of SBP.

## 4. Conclusions

This study investigated the effect of ultrasonic modification on the structural, intrinsic viscosity, color, thermal stability, and antioxidant capacity of SBP. Ultrasonic modification could increase the content of GalA and decrease the content of neutral sugars (Rha, Gal, Ara, Glc) in the SBP samples. The DE, intrinsic viscosity, *M_V_* and particle size of SBP decreased due to the partial degradation of its molecular chain. In addition, the color value of SBP also had obvious changes after ultrasonic treatments. FT-IR results further indicated that the glycosidic bonds of SBP were broken by ultrasound modification. The microstructure of modified SBP showed that ultrasonic treatment caused an increase in small molecular weight fragments of SBP samples. The thermal stability of SBP was obviously improved via ultrasonic treatment, which provided reference for further applications of SBP in thermal processing. The results of antioxidant activity analysis showed that ultrasonication was an effective method to improve the radical scavenging capacity and reducing power of SBP. All of the results in this study indicated that ultrasonic technology is an environmentally friendly, simple and effective way to improve the antioxidant capacity of SBP.

## Figures and Tables

**Figure 1 foods-12-01020-f001:**
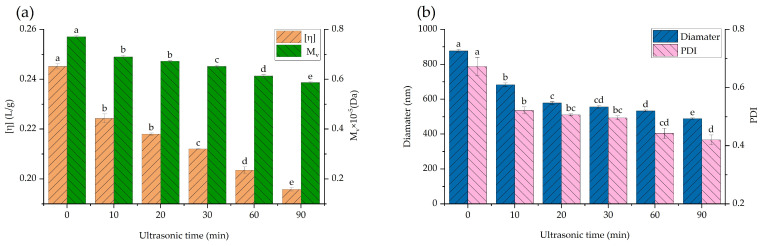
(**a**) Intrinsic viscosity and *M_V_* of native SBP and its ultrasonic degradation products. (**b**) The particle size and PDI of SBP and its ultrasonic degradation products. Different letters ^a–e^ indicate significant differences (*p* < 0.05).

**Figure 2 foods-12-01020-f002:**
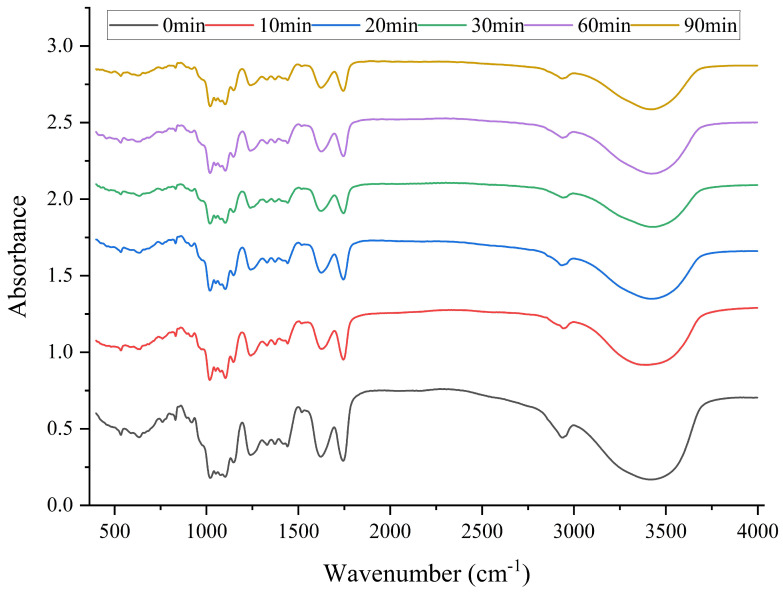
The FT-IR absorption peaks of different SBP samples.

**Figure 3 foods-12-01020-f003:**
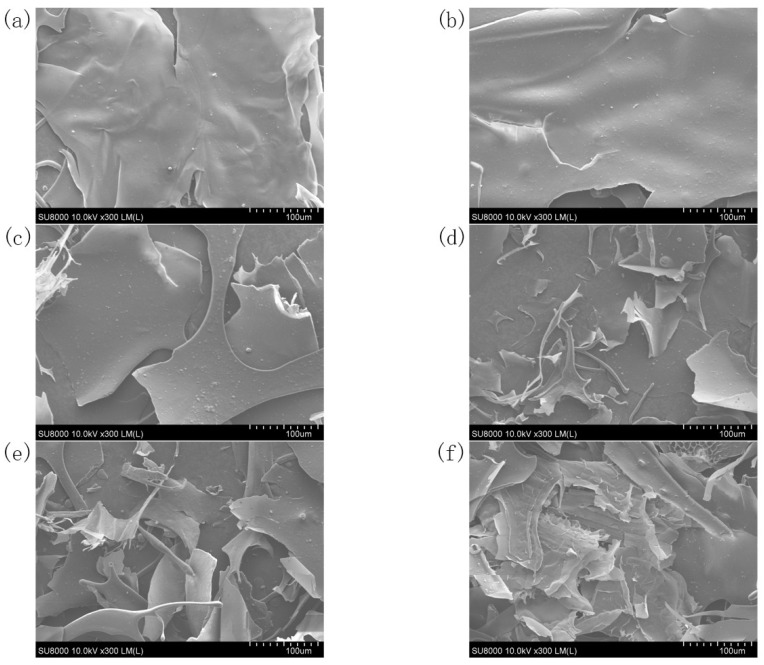
SEM image of native SBP and its ultrasonic degradation products. (**a**) Native, (**b**) 10 min, (**c**) 20 min, (**d**) 30 min, (**e**) 60 min, (**f**) 90 min.

**Figure 4 foods-12-01020-f004:**
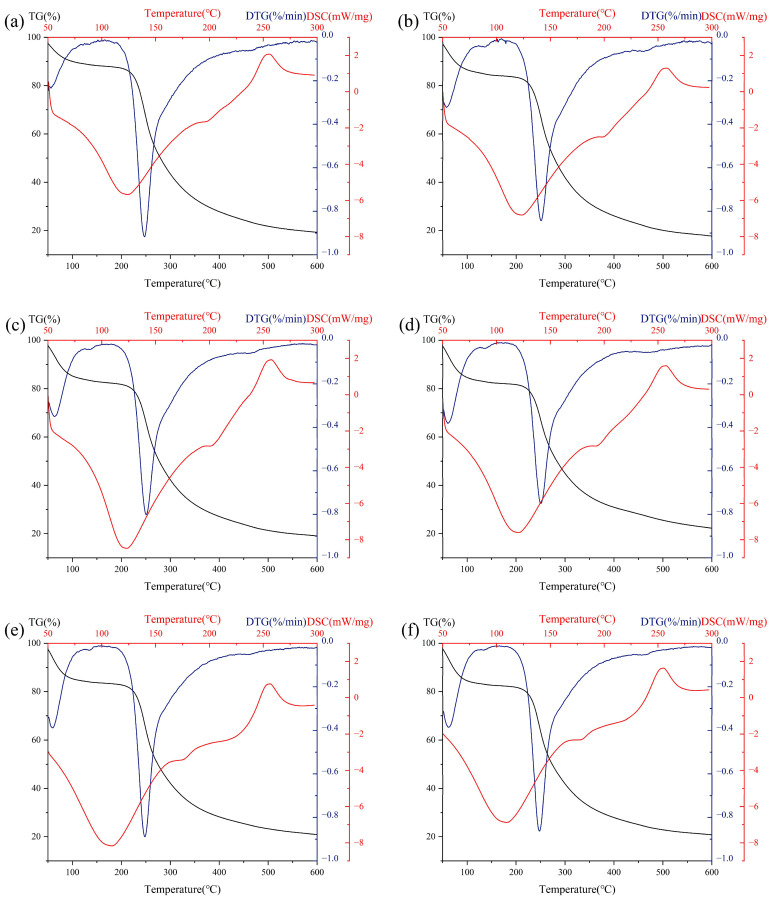
Thermal analysis of native SBP and its ultrasonic degradation products. (**a**) Native, (**b**) 10 min, (**c**) 20 min, (**d**) 30 min, (**e**) 60 min, (**f**) 90 min.

**Figure 5 foods-12-01020-f005:**
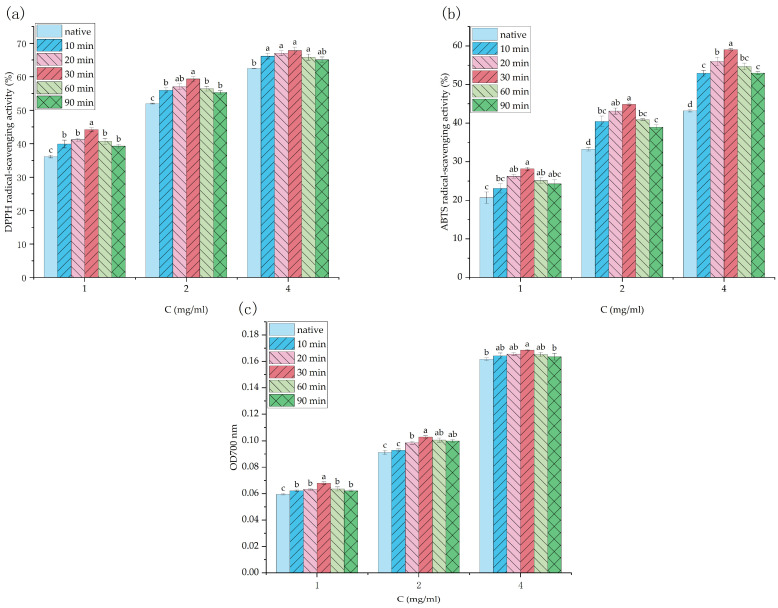
Antioxidant activity analysis of SBP and its ultrasonic degradation products. (**a**) DPPH radical scavenging activity, (**b**) ABTS radical scavenging activity and (**c**) reducing power. Different letters ^a–d^ indicate significant differences (*p* < 0.05).

**Table 1 foods-12-01020-t001:** Chemical composition of native SBP and its ultrasonic degradation products.

	Native	10 min	20 min	30 min	60 min	90 min
GalA (%)	63.68 ± 0.30 ^b^	65.19 ± 0.22 ^a,b^	66.33 ± 0.74 ^a,b^	66.76 ± 0.51 ^a,b^	68.28 ± 1.16 ^a^	68.15 ± 1.50 ^a^
Pro (%)	4.20 ± 0.14 ^a^	4.25 ± 0.04 ^a^	4.21 ± 0.13 ^a^	4.12 ± 0.12 ^a^	4.25 ± 0.02 ^a^	4.29 ± 0.08 ^a^
Gal (%)	8.53 ± 0.05 ^a^	8.4 ± 0.17 ^a,b^	8.17 ± 0.17 ^a,b,c^	7.91 ± 0.13 ^b,c,d^	7.64 ± 0.25 ^d^	7.63 ± 0.11 ^c,d^
Rha (%)	6.29 ± 0.03 ^a^	6.16 ± 0.07 ^a,b^	6.14 ± 0.06 ^a,b^	5.99 ± 0.14 ^b,c^	5.72 ± 0.04 ^c,d^	5.43 ± 0.06 ^d^
Ara (%)	3.16 ± 0.05 ^a^	2.69 ± 0.25 ^a,b^	2.47 ± 0.04 ^b,c^	2.41 ± 0.2 ^b,c^	2.14 ± 0.14 ^b,c^	2.08 ± 0.06 ^c^
Glc (%)	0.87 ± 0.02 ^a^	0.83 ± 0.02 ^a^	0.73 ± 0.01 ^b^	0.69 ± 0.01 ^b,c^	0.63 ± 0.02 ^c^	0.52 ± 0.02 ^d^
DE (%)	51.40 ± 0.99 ^a^	49.75 ± 0.21 ^a,b^	47.70 ± 1.56 ^b,c^	44.90 ± 0.57 ^c,d^	42.78 ± 0.39 ^d,e^	41.50 ± 0.71 ^e^

Different letters ^a−d^ indicate significant differences (*p* < 0.05).

**Table 2 foods-12-01020-t002:** Color values of native SBP and its ultrasonic degradation products.

	Native	10 min	20 min	30 min	60 min	90 min
L*	84.90 ± 0.78 ^c^	84.18 ± 0.20 ^c^	86.38 ± 0.12 ^b^	87.07 ± 0.34 ^a,b^	87.04 ± 0.39 ^a,b^	87.6 ± 0.13 ^a^
a*	0.67 ± 0.06 ^a^	0.62 ± 0.06 ^a^	0.41 ± 0.02 ^b^	0.34 ± 0.02 ^b^	0.15 ± 0.02 ^c^	0.02 ± 0.01 ^d^
b*	10.31 ± 0.34 ^a^	10.46 ± 0.14 ^a^	10.19 ± 0.05 ^a^	10.28 ± 0.02 ^a^	9.72 ± 0.12 ^b^	9.7 ± 0.02 ^b^

Different letters ^a–d^ indicate significant differences (*p* < 0.05).

**Table 3 foods-12-01020-t003:** Thermal analysis of native SBP and its ultrasonic degradation products.

	Native	10 min	20 min	30 min	60 min	90 min
$T_{m\;\mathrm{DTG}}$(°C)	247.67	251.17	250.67	250.67	248.00	247.83
$T_{d\;\mathrm{DSC}}$(°C)	124.17	123.43	123.23	120.39	109.12	109.32
$T_{m\;\mathrm{DSC}}$(°C)	255.63	258.02	257.55	257.24	256.38	255.12

## Data Availability

The data are available from the corresponding author.

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
