# Peer review of "Influences of Ultrasonic Treatments on the Structure and Antioxidant Properties of Sugar Beet Pectin"

_foods, 2023, doi:10.3390/foods12051020_

Round 1
Reviewer 1 Report (Previous Reviewer 2)
When the authors insert the following minor changes, I suggest the acceptance of the manuscript:
lines 39-41: I suggest that the sentence: „Pectin samples extracted from Opuntia robusta via different methods presented different that the pectin with lower molecular weight showed better antioxidant activity.“ should be corrected as follows: „Pectin samples extracted from Opuntia robusta via different methods presented different than the pectin with lower molecular weight which showed better antioxidant activity.“ Also please explain to what the word „different“ (after presented) is reffered to? It is still not clear. The native English speaker should revise it.
lines 109-117: Please revise the text under the 2.4 DE subtitle in order to make it understandable and clear, because it is still not clear. The native English speaker should revise it.
line 139: the sentence: „According to the previous study with slight modified [29].“ sholud also be revised by the native English speaker.
Author Response
Dear Editor and Reviewers,
With this letter we are submitting the revised version of our manuscript entitled ‘‘Influences of ultrasonic treatments on the structure and antioxidant properties of sugar beet pectin’’(Original title: Structural characterization and antioxidant properties of sugar beet pectin and its ultrasonic degradation products) by Yingjie Xu et al (Manuscript ID: foods-2218534). First of all, thanks for your trust once again and providing us with this opportunity to submit the revised version of our manuscript. We appreciate the detailed and constructive comments provided by the reviewers. We have carefully revised the manuscript by incorporating all the suggestions by the review panel and the manuscript was marked up using the "Track Changes” function (highlighted in red).

Reviewer 2 Report (Previous Reviewer 1)
Dear authors, great job indeed after first revision.
I have some comments and corrections via Track changes in word file.
Therefore,
please find enclosed word file.

Author Response
Dear Editor and Reviewers,
With this letter we are submitting the revised version of our manuscript entitled ‘‘Influences of ultrasonic treatments on the structure and antioxidant properties of sugar beet pectin’’(Original title: Structural characterization and antioxidant properties of sugar beet pectin and its ultrasonic degradation products) by Yingjie Xu et al (Manuscript ID: foods-2218534). First of all, thanks for your trust once again and providing us with this opportunity to submit the revised version of our manuscript. We appreciate the detailed and constructive comments provided by the reviewers. We have carefully revised the manuscript by incorporating all the suggestions by the review panel and the manuscript was marked up using the "Track Changes” function (highlighted in red).

Reviewer 3 Report (New Reviewer)
It seems that the manuscript was peer-reviewed and it is improved after revision. In my opinion, most of the parts are acceptable after the previous revision. There are a few comments as follows and need to address by the authors before the final decision by the editor.
1- The title is not clear. You need to make it more informative. A research title should show the "Why" and "How" of the research. (Not meaning to bring "Why" and "How" in the title!!).
2- Abstract: Improve the results in the abstract by incorporating of some quantitative data.
3- Keywords: choose some simple words as keywords.
4- Figure 1(b) missed significant letters for PDI.
5- References: Better to replace outdated references before 2020 if possible.
Author Response
Dear Editor and Reviewers,
With this letter we are submitting the revised version of our manuscript entitled ‘‘Influences of ultrasonic treatments on the structure and antioxidant properties of sugar beet pectin’’(Original title: Structural characterization and antioxidant properties of sugar beet pectin and its ultrasonic degradation products) by Yingjie Xu et al (Manuscript ID: foods-2218534). First of all, thanks for your trust once again and providing us with this opportunity to submit the revised version of our manuscript. We appreciate the detailed and constructive comments provided by the reviewers. We have carefully revised the manuscript by incorporating all the suggestions by the review panel and the manuscript was marked up using the "Track Changes” function (highlighted in red).

This manuscript is a resubmission of an earlier submission. The following is a list of the peer review reports and author responses from that submission.
Round 1
Reviewer 1 Report
I have critically and carefully evaluated this research article entitled as: “Structural characterization and antioxidant properties of sugar beet pectin and its ultrasonic degradation products” and I have reached the
following conclusions:
Authors, I recommend MINOR REVISION of this article in MDPI Journal Foods after bellow mentioned suggestions. Also, it is important to note that the article is well written and authors used appropriate literature.
My comments:
Line 39: O. robusta, you must use full Latin name, because You use this only on this place, therefore you should write as: Opuntia robusta
Line 91: You started with sentence as: According to the previous study [23]. It is not finished or?
Line 92: m-hydroxydiphenyl method (m with itac font)
Line 109: you ommied space between 200 and used unit (please if you used micro L write as μL) not uL and the same comment is for the Line 113
Line 169: which volume of DPPH reagent you used?
Line 177: please use unit in the whole text, for example in the ABTS assay 5 mL not with a words
Line 181: please check through whole text of the MS and correct this: you must used space between number and appropriate unit (for example 2 mL)
Line 192: FeCl3
Lines 201 and 202: second part of the title you replace below the Table 1 (Different letters ad indicated significant differences at the 0.05 level (P<0.05))
All letters which you write with numbers in the Table 1 must write with superscript
Also when you write for exampe ab please rewrite as a,b with superscript
Line 261: the same comment as for table 1, part of the title please replace below the Table 2 (Different letters ad indicated significant differences at the 0.05 level (P<0.05), and the all numbers write in superscript
Line 390: where is Author contributions?
References:
please check whole reference list. It is not well cited.
For example:
Ref 1: Mehrlander, K.; Dietrich, H.; Sembries, S.; Dongowski, G.; Will, F. Structural characterization of oligosaccharides and polysaccharides from apple juices produced by enzymatic pomace liquefaction. J. Agric. Food. Chem[M1] . [M2] 2002[M3] , 50[M4] ,1230-1236. doi:10.1021/jf011007i.
Please correct other references based on the corrected Reference 1!
Line 431: You ommited space in the reference number 12: Fruit Peel of Opuntia robusta.
Line 517: please delete number 48 which you write with authos name: 48.Sun
Line 524525: please use abbreviation of the Journal name
[M1]Between Journal abbr and year of publication you must use space
[M2]All journal names must be written in italic and you must use abbreviations of Journal name
[M3]Year of publication is in bold
[M4]Number of volume is in italic

Author Response
Cover Letter
Dear Editor and Reviewers,
With this letter we are submitting the revised version of our manuscript entitled ‘‘Structural characterization and antioxidant properties of sugar beet pectin and its ultrasonic degradation products’’ by Yingjie Xu et al (Manuscript ID: foods-2105812). First of all, thanks for providing us with this great opportunity to submit a revised version of our manuscript. We appreciate the detailed and constructive comments provided by the reviewers. We have carefully revised the manuscript by incorporating all the suggestions by the review panel and are highlighted in red.
Reviewer 1:
Comments to the Author
- robusta, you must use full Latin name, because You use this only on this place, therefore you should write as: Opuntia robusta.
Reply: We sincerely thank the reviewer for careful reading. As suggested by the reviewer, we have corrected the “O. robusta” into “Opuntia robusta”. (Line 39)
Previous study has shown that pectin samples extracted from Opuntia robusta via different methods presented different that the pectin with lower molecular weight showed better antioxidant activity. A possible explanation for this phenomenon is that the pectin with lower molecular weight is favorable for exposing more active sites of antioxidant activity [12]
- You started with sentence as: According to the previous study [23]. It is not finished or?
Reply: Thanks for the reminder, the misunderstanding due to my expression problem has been modified. (Line 92-93)
The GalA content of different SBP samples (with or without ultrasonic treatment) was determined by m-hydroxydiphenyl method [23].
- m-hydroxydiphenyl method (m with itac font)
Reply: Thanks a lot for the very careful comment and apologize for the mistake caused by our carelessness. We recheck the manuscript and revise it. (Line 93)
- you ommied space between 200 and used unit (please if you used micro L write as μL) not uL and the same comment is for the Line 113.
Reply: Thanks. We have corrected these mistakes based on your suggestions. (Line 114)
- which volume of DPPH reagent you used?
Reply: The volume of DPPH reagent which we used is 2 mL. Thanks for the reminder, we made corrections in the manuscript.
- please use unit in the whole text, for example in the ABTS assay 5 mL not with a words.
Reply: Thanks. We have corrected these mistakes based on your suggestions. (Line 179)
- please check through whole text of the MS and correct this: you must used space between number and appropriate unit (for example 2 mL)
Reply: Thanks for your kind reminding. We recheck the reference format and revise it in the revised manuscript.
- Line 192: FeCl3
Reply: We feel sorry for our carelessness. In our resubmitted manuscript, the typo is revised. Thanks for your correction. (Line 194)
- All letters which you write with numbers in the Table 1 and Table 2 must write with superscript. Also when you write for example ab please rewrite as a,bwith superscript
Reply: Thanks for your great suggestion on improving the accessibility of our manuscript. we have corrected this mistake. (Line 238 239, 277 278)
- Line 390: where is Author contributions?
Reply: Thanks for your great suggestion on our manuscript. We have added Author contributions. (Author Contributions: Conceptualization, Investigation, Formal analysis, Validation, Visualization, Writing-original draft-Y.X.; Investigation, Validation, Funding acquisi-tion, Resources, Supervision—J.Z.; Conceptualization, Investigation, Validation, Writ-ing—review and editing, Funding acquisition, Resources, Supervision—X.G.; Concep-tualization, Validation, Formal analysis—J.H. and T.L. All authors have read and agreed to the published version of the manuscript.)
- References: please check whole reference list. It is not well cited.
Reply: we were really sorry for our careless mistakes and correct it. Thank you for your reminding.
- Please correct other references based on the corrected Reference 1!
Line 431: You ommited space in the reference number 12: Fruit Peel of Opuntia robusta.
Line 517: please delete number 48 which you write with authos name: 48.Sun
Line 524525: please use abbreviation of the Journal name
Reply: Thanks for your careful checks. We are sorry for our carelessness. Based on your comments, we have corrected the references. (Line 440 523 530)
To sum up, we have rechecked the full text and try our best to revise it on the basis of comments. We hope this revised manuscript has addressed your concerns. Please do not hesitate to contact us at the address below if there are any questions.
Tel: +86-18709936830
Email: 20202111023@stu.shzu.edu.cn
Reviewer 2 Report
This manuscript contains comprehensive overview of structural and antioxidant characteristics of sugar beet pectin and its ultrasonic degradation products.
I suggest a minor revision.
Specific remarks are as follows:
Throughout the manuscript, latin terms should be written in italic
Line 7: the word „major“ should be deleted
Lines 39-41: the sentence „Pectin is also a biologically active polysaccharide. Pectin samples extracted from O. robusta via different methods presented different that the pectin with lower molecular weight showed better antioxidant activity.“ is not clear. Please revise it.
Line 55: the word „structure“ should be changed to „structural“
Line 58: the word „inhibited“ should be changed to „inhibit“
Line 71: the word „condition“ should be changed to „conditions“
Line 91: a comma should be placed after [23] and „The“ should be changed to „the“
Line 92: m- should be written in italic
Line 98: a comma should be placed after NaOH and „And“ should be changed to „and“
Line 99: a comma should be placed after „After that“
Please revise the text under the 2.4 DE subtitle in order to make it understandable and clear.
The equation numbers should be cited in the text of the manuscript, i. e. they need to be addressed.
Line 111: V1 – 1 should be written in index
Line 113: -COOCH3 – 3 should be written in index
Line 115: V2 – 2 should be written in index
Line 116: % after 100 should be omitted; this applies to equations 5 and 6, as well
Molecular weight should be shortened as MW and not MV, thorought the manuscript
Line 121: t_s should be written as ts (s is in index); this applies to: t_0, η_r, η_sp, A_i, A_j, A_0, A_x, T_(m DSC), T_(d DTG), T_(d DSC)
Line 130: the word „weightt“ should be changed to „weight“
Line 134: „10-5“ should be written in the exponent
Line 136: „sample was“ should be changed to „samples were“
Line 138: a comma should be placed after „rpm“, „And“ should be changed to „and“ and „sample“ should be changed to „samples“
The appropriate reference for the text under the 2.6., 2.7., 2.8. and 2.9. subtitles should be given.
Line 144: a comma should be placed after the word „measurement“ and „And“ should be changed to „and“
Line 152: cm−1 should be written with -1 in superscript; this applies to the rest of the text, as well
Line 189: K3[Fe (CN)6] – 3 and 6 should be written in index
Line 192: FeCl3 - 3 should be written in index
Line 196: „are“ should be changed to „were“
Line 203: „shows“ should be changed to „show“
Line 207: „68.15%“ should be changed to „68.28%“
Line 213: „occured“ should be deleted
Line 220: „those“ should be changed to „this“
Line 283: „attribute“ should be changed to „attributed“
Throughout the manuscript, the tables and figures should be announced in the text and then represented. For example, lines 201 and 202 and the approptiate table shoud be given after the line 213. This applies to all tables anf figures in the manuscript.
Line 307: „the“ should be placed before „thermal“
Line 308: „it“ should be added between „As“ and „can“
Line 314: „Table 4“ should be changed to „Table 3“
Line 320: „degradation“ should be changed to „degraded“
Line 322: „positive correlation“ should be changed to „positively correlated“
Line 323: „decreased“ should be changed to „decrease“
Line 326: a dot after „enhanced“ should be deleted
Line 328: the word „obviously“ should be deleted
Line 338: DPPH~ - the sign ~ should be deleted
Line 349: „IC50“ should be changed to „IC50“
Line 380: „broke“ should be changed to „broken“
Figure 5c - % is not the measurement unit for reducing power
Throughout the manuscript, a sentence should not start with a numerical value and a space should be placed between the number and the appropriate measurement unit.
Equations 5 and 6 should be numbered as 6 and 7.
P for statistical significance should be written as p.
The letters for statistical significance in the tables should be written in superscript.
Author Response
Cover Letter
Dear Editor and Reviewers,
With this letter we are submitting the revised version of our manuscript entitled ‘‘Structural characterization and antioxidant properties of sugar beet pectin and its ultrasonic degradation products’’ by Yingjie Xu et al (Manuscript ID: foods-2105812). First of all, thanks for providing us with this great opportunity to submit a revised version of our manuscript. We appreciate the detailed and constructive comments provided by the reviewers. We have carefully revised the manuscript by incorporating all the suggestions by the review panel and are highlighted in red.
Reviewer 2:
Comments to the Author
- the word „major“ should be deleted
Reply: (Line 7) Thanks for your careful checks.We have deleted it.
- the sentence „Pectin is also a biologically active polysaccharide. Pectin samples extracted from O. robusta via different methods presented different that the pectin with lower molecular weight showed better antioxidant activity.“ is not clear. Please revise it.
Reply: (Line 38 41) We have re-written this part according to the Reviewer’s suggestion.
Pectin not only has gel properties, but also has antioxidant capacity. Previous study has shown that pectin samples extracted from Opuntia robusta via different methods presented different that the pectin with lower molecular weight showed better antioxidant activity.
- the word „structure“ should be changed to „structural“;
the word „inhibited“ should be changed to „inhibit“;
the word „condition“ should be changed to „conditions“
Reply:(Line 55 58 73) We sincerely appreciate the valuable comments. We have changed the word “structure” to “structural”; “inhibited” to “inhibit”; “condition” to “conditions”
- a comma should be placed after [23] and „The“ should be changed to „the“
Reply: (Line 93 94) Because the expression here is not clear enough, I have revised it.
The GalA content of different SBP samples (with or without ultrasonic treatment) was determined by m-hydroxydiphenyl method [23].
- m- should be written in italic;
a comma should be placed after NaOH and „And“ should be changed to „and“;
a comma should be placed after „After that“
Reply: (Line 94 100 101) Thank you for your careful inspection. We have made corrections.
- Please revise the text under the 2.4 DE subtitle in order to make it understandable and clear.
Reply: (Line 109) Thank you for your advice. We have changed it into “Esterification degree (DE)”.
- The equation numbers should be cited in the text of the manuscript, i. e. they need to be addressed.
Reply: (Line 175 186) Thanks a lot for your excellent advice. We will pay attention to this problem in the future. Because the previous research did not quote the equation in the text, we also think that it is not necessary to quote the equation.
- Ogutu, F.O.; Mu, T.H. Ultrasonic degradation of sweet potato pectin and its antioxidant activity. Ultrason. Sonochem. 2017, 38,726-734. doi:10.1016/j.ultsonch. 2016.08.014.
- Chen, X.; Qi, Y.; Zhu, C.; Wang, Q. Effect of ultrasound on the properties and antioxidant activity of hawthorn pectin. Int. J. Biol. Macromol. 2019, 131,273-281. doi:10.1016/j.ijbiomac. 2019.03.077.
- V1 – 1 should be written in index;
- -COOCH3 – 3 should be written in index;
- V2 – 2 should be written in index;
- % after 100 should be omitted; this applies to equations 5 and 6, as well
Reply: (Line 113-119) Thank you for your advice. We have made corrections in the manuscript.
- Molecular weight should be shortened as MWand not MV, thorought the manuscript
Reply: (Line 136) We sincerely appreciate the valuable comments. Because the molecular weight is calculated by Mark-Houwink equation, the viscosity average molecular weight is obtained, so it is expressed by MV. MV is one of the common methods to express the molecular weight of polymers.
10) t_s should be written as ts (s is in index); this applies to: t_0, η_r, η_sp, A_i, A_j, A_0, A_x, T_(m DSC), T_(d DTG), T_(d DSC)
Reply: Thank you for your advice. We have made corrections.
- the word „weightt“ should be changed to „weight“;
- „10-5“ should be written in the exponent;
- „sample was“ should be changed to „samples were“;
- a comma should be placed after „rpm“, „And“ should be changed to „and“ and „sample“ should be changed to „samples“
Reply: (Line 133 137 142 )Thank you for the detailed review. We have carefully and thoroughly proofread the manuscript to correct all the grammar and typos.
- The appropriate reference for the text under the 2.6., 2.7., 2.8. and 2.9. subtitles should be given.
Reply: Thanks a lot for your reminding. 2.6 subtitles add relative references as Ref 29, but FT-IR, SEM and Color analysis are very routine testing methods, so no references are added.
29.Chen, X.; Qi, Y.; Zhu, C.; Wang, Q. Effect of ultrasound on the properties and antioxidant activity of hawthorn pectin. Int. J. Biol. Macromol. 2019, 131,273-281. doi:10.1016/j.ijbiomac.2019.03.077.
- a comma should be placed after the word „measurement“ and „And“ should be changed to „and“;
- cm−1 should be written with -1 in superscript;
- this applies to the rest of the text, as well;
- K3[Fe (CN)6] – 3 and 6 should be written in index;
- FeCl3 - 3 should be written in index;
- „are“ should be changed to „were“; „shows“ should be changed to „show“; „68.15%“ should be changed to „68.28%“;
- „occured“ should be deleted;
- „those“ should be changed to „this“;
- „attribute“ should be changed to „attributed“
Reply: (Line 147 155 192 195 199 204 210 217 224 289)We really appreciate your great patience and careful. We feel sorry for our carelessness. In our resubmitted manuscript, the typo is revised.
- Throughout the manuscript, the tables and figures should be announced in the text and then represented. For example, lines 201 and 202 and the approptiate table shoud be given after the line 213. This applies to all tables andfigures in the manuscript.
Reply:Thank you for your advice. The tables and figures have announced in the text and then represented.
- „the“ should be placed before „ thermal“
Reply: (Line 311) Thank you for your advice. “Thermal analysis technology is often used to study thermal the stability and intermolecular interaction of macromolecules”. This sentence was deleted according to the request of Reviewer 4.
- „it“should be added between „As“and „can“
- „Table 4“should be changed to „Table 3“
- „degradation“ should be changed to„ degraded“
Reply: Thank you for your advice. We made corrections in the manuscript.
(Line 311)„it“ have added between „As“ and „can“
(Line 337)„Table 4“ have changed to „Table 3“
(Line 323)„degradation“ have changed to „degraded“
- „positive correlation“ should be changed to „positively correlated“
Reply: (Line 323-325) Thank you for your advice. According to the suggestion of reviewer 3, we have revised this sentence.
Previous studies have shown that the higher content of GalA in pectin means that more energy is needed to complete the transformation [43].
- „decreased“ should be changed to „decrease“
Reply: ( Line 325) Thank you for your advice. We have changed “decreased” to “decrease”
- a dot after „enhanced“ should be deleted
- the word „obviously“ should be deleted
Reply: ( Line 328-333) Thank you for your advice. According to the suggestion of reviewer 4, we have revised this sentence.
In this study, the SBP after ultrasonic treatment was degraded, the content of GalA increased relatively and the molecular weight decreased, but the thermal stability was higher than that of the original pectin. This may be because ultrasonic treatment changed the microstructure of SBP. Above results indicated that the ultrasonic treatment could improve the thermal stability of SBP.
- DPPH - the sign ~ should be deleted~
Reply: Thank you for your suggestion. We have deleted it.
- „IC50“ should be changed to „IC50“
Reply: (Line 354)Thank you for your suggestion. We have corrected it.
- „broke“ should be changed to „broken“
Reply: (Line 389)Thank you for your suggestion. We have corrected it.
- Figure 5c - % is not the measurement unit for reducing power
Reply: Thank you for your suggestion. We have corrected it.
- Throughout the manuscript, a sentence should not start with a numerical value and a space should be placed between the number and the appropriate measurement unit.
Reply: (Line 111-112 171-173 180-181 194-195)Thank you for your suggestion. We have revised it.
- Equations 5 and 6 should be numbered as 6 and 7.
Reply: (Line 175 186 )Thank you for your suggestion. We have corrected it.
- P for statistical significance should be written as p.
Reply: Thank you for your suggestion. We have corrected it.
- The letters for statistical significance in the tables should be written in superscript.
Reply: We were really sorry for our careless mistakes. Thank you for your reminder. We checked and corrected it in the manuscript.
To sum up, we have rechecked the full text and try our best to revise it on the basis of comments. We hope this revised manuscript has addressed your concerns. Please do not hesitate to contact us at the address below if there are any questions.
Tel: +86-18709936830
Email: 20202111023@stu.shzu.edu.cn
Reviewer 3 Report
The manuscript "Structural characterization and antioxidant properties of sugar beet pectin and its ultrasonic degradation products" presents a comprehensive and well-organized study on the effect of ultrasonic treatment on the properties of sugar beet pectin. However, some statements such as the relationship of ultrasonication time with antioxidant activity are not clear, particularly with the statement that the best treatment time is 30 min. Some additional analyzes could be performed to support such results, for example, gelling tests or determination of uronic acid content. On the other hand, a review of the wording is recommended in the methodology section. In addition, some modifications are suggested in the checklist below.
General observations:
Please define abbreviations, for example, DE (Line 104)
Line 22. “increased” is correct?
Line 51. “was” for “were”
Line 54. Please use italics for scientific names, attend this indication throughout the manuscript. Example: Porphyra yezoensis
Line 70. Please delete “by us”.
Line 88. previous study [23], the GalA content.
Line 105-112. Improve the wording
Line 127. weight for weight
Line 235. Equation 5?
Author Response
Cover Letter
Dear Editor and Reviewers,
With this letter we are submitting the revised version of our manuscript entitled ‘‘Structural characterization and antioxidant properties of sugar beet pectin and its ultrasonic degradation products’’ by Yingjie Xu et al (Manuscript ID: foods-2105812). First of all, thanks for providing us with this great opportunity to submit a revised version of our manuscript. We appreciate the detailed and constructive comments provided by the reviewers. We have carefully revised the manuscript by incorporating all the suggestions by the review panel and are highlighted in red.
Reviewer 3:
Comments to the Author
- (Line 14) We may not express it well enough. Under the conditions of this study, compared with other treatment groups, the antioxidant activity of 30min is the best. In order to be rigorous in research, we have revised the manuscript.
- “increased” is correct?
Reply: (Line 24-26) Thank you for your advice. According to the suggestion of reviewer 4, we have revised this sentence.
Due to the complexity of the fine structure domain of pectin, the difference of neutral sugar type and content affects the functional characteristics of pectin [2,3]
2)“was” for “were”
Reply: (Line 54) We were really sorry for our careless mistakes. Thank you for your reminder.
3) Please use italics for scientific names, attend this indication throughout the manuscript. Example: Porphyra yezoensis
Reply: (Line 57) Thank you for your careful inspection. We have made this correction.
4) Please delete “by us”.
Reply: (Line 76) Thank you for your careful inspection. We have made this correction.
- previous study [23], the GalA content.
Reply: (Line 76) Thank you for your careful inspection. The expression here is not clear enough, I have revised it.
The GalA content of different SBP samples (with or without ultrasonic treatment) was determined by m-hydroxydiphenyl method [23].
- Improve the wording!
Reply: (Line 110-117) Thank you for your careful inspection. The expression here is not clear enough, I have revised it.
According to the previous study with slight modified [25]. The SBP samples were all prepared at 1 mg/mL, the conical flask containing above solution 20 mL and added 200 μL of phenolphthalein indicator. Titrated with 0.05 M NaOH until discoloration occurred, the consumption volume was recorded as V1. Then added 10 mL of 0.1 M NaOH, stirred continuously for 30 min to saponify SBP. And the excess NaOH was neutralized by using 10 mL of 0.1M HCl. At this time, the -COOCH3 in SBP was converted to -COOH. Then added 200 μL of phenolphthalein indicator to the solution and titrated the solution with 0.05 M NaOH until the solution turned pink, the consumption volume was recorded as V2.
- weight for weight
Reply: (Line 133) Thank you for your careful inspection. I have revised it.
- Equation 5?
Reply: Thank you for your reminding. Equation (5) is Mark-Houwink equation, the functional relationship between the intrinsic viscosity of dilute solution of polymers and the viscosity average molecular weight.
To sum up, we have rechecked the full text and try our best to revise it on the basis of comments. We hope this revised manuscript has addressed your concerns. Please do not hesitate to contact us at the address below if there are any questions.
Tel: +86-18709936830
Email: 20202111023@stu.shzu.edu.cn
Reviewer 4 Report
These are my main comments on the manuscript (foods-2105812) entitled “Structural characterization and antioxidant properties of sugar beet pectin and its ultrasonic degradation products”. Following moderate revisions should be incorporated in the manuscript prior to acceptance.
1. I have concerns about the manuscript sections that I believe need to be addressed in order to improve its clarity.
2. A hypothesis for this work is needed.
3. In results, Statistical methods are missing in methods section. The authors must present the values (F-values, degree freedom, p-value, etc.) obtained in each statistical analysis.
4. Results and discussion should be divided in two sections.
5. Other revisions could be checked in PDF attached.

Author Response
Cover Letter
Dear Editor and Reviewers,
With this letter we are submitting the revised version of our manuscript entitled ‘‘Structural characterization and antioxidant properties of sugar beet pectin and its ultrasonic degradation products’’ by Yingjie Xu et al (Manuscript ID: foods-2105812). First of all, thanks for providing us with this great opportunity to submit a revised version of our manuscript. We appreciate the detailed and constructive comments provided by the reviewers. We have carefully revised the manuscript by incorporating all the suggestions by the review panel and are highlighted in red.
Reviewer 4:
Comments to the Author
- Keywords should be in alphabetic order. Also, keywords serve to widen the opportunity to be retrieved from a database. To put words that already are into title and abstracts makes KW not useful. Please choose terms.
Reply: (Line 18) We think this is an excellent suggestion. We have revised it.
Keywords: Sugar beet pectin; Ultrasonic modification; Intrinsic viscosity; thermal stability; Antioxidant capacity
- Revise this sentence to eliminate rewordiness (The type and number of NS chains increased that complexity of the structure of pectin [2], because differences in the fine structure domain led to functional differences of pectin [3].)
Reply: (Line 24-26) We think this is an excellent suggestion. We have revised it.
Due to the complexity of the fine structure domain of pectin, the difference of neutral sugar type and content affects the functional characteristics of pectin [2,3].
- Again, revise this sentence to eliminate rewordiness. (Therefore, the source of raw materials [8,9], and the different structural domains of pectin lead to different functional characteristics and applications.)
Reply: (Line 28-30) We have re-written this part according to the Reviewer’s suggestion.
In addition, the source of raw materials and different extraction methods of pectin also lead to different functional characteristics and applications [8,9].
- Again, revise this sentence to eliminate rewordiness.(Pectin with different degrees of esterification can be selected according to the different application scenarios.)
Reply: (Line 32-33) We have re-written this part according to the Reviewer’s suggestion
Pectin with different degrees of esterification has different applications.
- O. robusta
Reply: (Line 39) Thank you for your advice. We have changed “O. robusta
” to “Opuntia robusta”
- “Porphyra yezoensis”in italic
Reply: (Line 57) Thank you for your advice. We have changed “Porphyra yezoensis
” to “Porphyra yezoensis”
- Also, a hypothesis for this study is needed.
Reply: (Line 70-72) We have re-written this part according to the Reviewer’s suggestion
- After hydrolysis
Reply: (Line 98) Thank you for your advice. We have changed “After hydrolysis” to “Then”
- Duncan's multiple range test is not rigorous, I suggest replace by Tukey HSD test
Reply: (Line 199-201) The relevant research literature uses Duncan’s test, we also use Duncan’s test, and we agree with your suggestion. We will improve the data processing in the future.
- Qiu, W.Y.; Cai, W.D.; Wang, M.; Yan, J.K. Effect of ultrasonic intensity on the conformational changes in citrus pectin under ultrasonic processing. Food Chem. 2019, 297, 125021. doi:10.1016/j.foodchem.2019.125021.
- Chen, X.; Qi, Y.; Zhu, C.; Wang, Q. Effect of ultrasound on the properties and antioxidant activity of hawthorn pectin. Int. J. Biol. Macromol. 2019, 131,273-281. doi:10.1016/j.ijbiomac.2019.03.077.
- In table, provide the F value, degree freedom, and p-value obtained from ANOVA
Reply: The relevant research literature does not provide these data, we did not display them, and the data is complicated and inconvenient to display. We also agree with your suggestion, and we will improve the data processing in the future.
- Ogutu, F.O.; Mu, T.H. Ultrasonic degradation of sweet potato pectin and its antioxidant activity. Ultrason. Sonochem. 2017, 38,726-734. doi:10.1016/j.ultsonch.2016.08.014.
- Chen, X.; Qi, Y.; Zhu, C.; Wang, Q. Effect of ultrasound on the properties and antioxidant activity of hawthorn pectin. Int. J. Biol. Macromol. 2019, 131,273-281. doi:10.1016/j.ijbiomac.2019.03.077.
- This information should be in discussion section (which was similar to Wang et al. [33].)
Reply: Thanks a lot for your reminding. We have re-written this part according to the Reviewer’s suggestion.
but the content of NS was affected, which was similar to the results of Wang et al. They found that ultrasonic treatment had no significant effect on monosaccharide types of yellow tea polysaccharide [33].
- Again, this information should be in discussion section. (In the ultrasonic-assisted extraction of pectin from mango peel, the content of GalA was also positively correlated with the ultrasonic intensity within a certain range [34].)
Reply: Thanks a lot for your reminding. We have re-written this part according to the Reviewer’s suggestion.
The same phenomenon occurred in the process of ultrasonic-assisted extraction of pectin from mango peel. With the increase of ultrasonic power, the content of GalA in pectin extracted from mango peel increased significantly. Ultrasonic wave may lead to the degradation of pectin side chain, thus increasing the GalA content of pectin [34]
- Again, this information should be in discussion section. (A possible explanation for those phenomena was that the rigid semi-flexible chain of SBP molecule transformed to a flexible chain or even flexible curled chain after ultrasonic treatment which require higher ultrasonic intensity or other auxiliary methods to break [24].)
Reply: Thank you for your reminding. Because there is no data to prove the chain conformation of SBP, the rigidity of the chain discussed here is not rigorous enough, so we will discuss the modification and delete the references.
A possible explanation for this phenomenon was that the structure of SBP will not be further degraded under this condition, which require higher ultrasonic intensity or other auxiliary methods to break.
- Again, this information should be in discussion section. (DE is the ratio of the amount of esterified GalA to the total GalA content [35].)
Reply: Thank you for your reminding. We explained it again.
Pectin is mainly composed of GalA residues in which some of the C-6 carboxyl groups are methyl-esterified. DE is the ratio of the amount of esterified GalA to the total GalA content [35].
- Again, this information should be in discussion section. (Previous studies showed that pectin with low esterification degree contains more free carboxyl groups [36].)
Reply: Thank you for your reminding.Because the explanation is not rigorous enough, we restated it and quoted a more convincing document.
Previous studies have showed that pectin with low esterification degree contains more free carboxyl groups because ultrasonic hydrolysis ester groups, reducing esterification degree [20].
- (Line 244-246) How was it positively correlated? It is necessary to carry out a Pearson correlation to confirm it.
Reply: Thank you for your reminding. Mark-Houwink equation gives the relationship between the intrinsic viscosity of polymer solution and the molecular weight of polymer. For a polymer solution at a given temperature, within a certain range of molecular weight, k and α are constants independent of molecular weight.
Figure 1 (a) showed the intrinsic viscosity and MV at different ultrasonic treatment time.
- Again, this information should be in discussion section (Considering the pectin particle size was positively correlated with the molecular weight, the molecule of SBP had been obviously degraded by ultrasonic treatment [30])
Reply: Thanks a lot for your reminding. Pearson correlation analysis of pectin particle size and molecular weight proves that they are positively correlated.
Considering the pectin particle size was positively correlated with the molecular weight (r=0.962, p<0.05), the molecule of SBP had been obviously degraded by ultra-sonic treatment [29]
- Again, this information should be in discussion section (The value of pectin PDI would be positively correlated with its values of particle size and degree of polymerization [37].)
Reply: Thanks a lot for your reminding. We have revised it.
The polydispersity index (PDI) is an indicator of the degree of homogeneity in the reaction system [36]
- Again, this information should be in discussion section (In addition, the free radicals generated by ultrasound could cause the oxidation and degradation of chromogenic group which induced a color change in SBP [38].)
Reply: Thanks a lot for your reminding. We think that the previous references are not enough to explain the problem, so we reinterpret and quote new references.
Long's research showed that the free radical produced by H2O2-VC destroyed the glycosidic bond, and the brightness of degraded polysaccharide was improved [37].
- Again, this information should be in discussion section.The peaks at 2941 cm-1, 1441 cm-1, and 1102 cm-1 were the stretching vibrations of C-H, C-OH, and C-O, respectively [39]. The absorption peaks at 859 cm-1 and 515 cm-1 indicated the existence of α- and β- glycosidic bonds [40].
Reply: Thanks a lot for your reminding. These peaks are characteristic areas of pectin, and FT-IR can't explain the changes of these peaks, so we can only analyze them simply.
- Revise this sentence to eliminate rewordiness. (The increase of tiny fragments of SBP implied the increase of its specific surface area, and further indicated that the mechanical wave generated by ultrasonication degraded the pectin structure.)
Reply: Thanks a lot for your reminding. Because the previous analysis was not clear enough, we revised it.
The increase of tiny fragments of SBP further indicated that the mechanical wave generated by ultrasonication degraded the pectin structure.
- Again, this information should be in discussion section (Previous studies have shown that the GalA content of pectin was positive correlation with the energy required for its conversion [44]. The decreased in molecular weight also affectsthe thermal stability of SBP. Chen’s study showed that the molecular weight of polysaccharides after ultrasonic degradation decreased, the structure was loose, and the intermolecular hydrogen bonding was enhanced, and the thermal stability was also enhanced.[45])
Reply: Thanks a lot for the very careful comment. We have revised it.
Previous studies have shown that the higher content of GalA in pectin means that more energy is needed to complete the transformation [43]. The decrease in molecular weight also affects the thermal stability of SBP. Chen’s study showed that the molecu-lar weight of polysaccharides after ultrasonic degradation decreased, resulted in the structure was loose, the intermolecular hydrogen bonding was enhanced and thereby contributed to its high thermal stability [44]. In this study, the SBP after ultrasonic treatment was degraded, the content of GalA increased relatively and the molecular weight decreased, but the thermal stability was higher than that of the original pectin. This may be because ultrasonic treatment changed the microstructure of SBP. Above results indicated that the ultrasonic treatment could improve the thermal stability of SBP.
- (Line 375) For mean classification, letters should be in ascendant order to major value bar. Revise
Reply: Thanks for your kind reminding. We have revised it in the manuscript.
- (Line 354-355) Again, this information should be in discussion section (With regard to ABTS radical scavenging ability is concerned, low molecular weight pectin had a lower IC50 value and better antioxidant activity [45].)
Reply: Thank you for your constructive comment. We made a supplementary explanation.
With regard to ABTS radical scavenging ability is concerned, low molecular weight pectin had a lower IC50 value and better antioxidant activity [45]. It is similar to the result of improving ABTS radical scavenging capacity of degraded SBP in this study.
- (Line 364-374) Results and discussion should be divided in two sections (Previous studies also showed that pectin with lower molecular weight and higher GalA content would produce more reducing ends which were conducive to improving its reduction capacity [47,48]. Hydroxyl groups contained in polysaccharides have free radical scavenging ability [49]. A reduction in DE increased the number of free carboxyl groups upstream to the ultrasound modified SBP chain structure and the accessible of hydroxyl groups on the surface [50]. Moreover, the intrinsic viscosity decreased and the solubility increased, which was beneficial to the exposure of active parts of SBP and the improvement of antioxidant activity. A similar conclusion was reached by Venzon et al [51]. Therefore, facts proved that the ultrasonic modification is an effective way to significantly improve the reducing power of SBP.)
Reply: Thanks for your kind reminding. And we revised it.
Generally speaking, for a given concentration of SBP, too long or too short ultrasonic treatment condition was not conducive to improving the antioxidant capacity of SBP. When the ultrasonic treatment time was too long, the active structures of SBP could be damaged by ultrasonic wave. But if the ultrasonic treatment time was insufficient, the antioxidant active sites of SBP cannot be fully exposed. A reduction in DE increased the number of free carboxyl groups upstream to the ultrasound modified SBP chain structure and the accessible of hydroxyl groups on the surface [49]. Moreover, the intrinsic viscosity decreased, and the solubility increased, which was beneficial to the exposure of active parts of SBP and the improvement of antioxidant activity. A similar conclusion was reached by Venzon et al. [50]. Therefore, facts proved that the ultrasonic modification is an effective way to significantly improve the reducing power of SBP.
To sum up, we have rechecked the full text and try our best to revise it on the basis of comments. We hope this revised manuscript has addressed your concerns. Please do not hesitate to contact us at the address below if there are any questions.
Tel: +86-18709936830
